# Privacy-Preserving Split Learning with Vision Transformers using Patch-Wise Random and Noisy CutMix

**Seungeun Oh**  *seoh@ramo.yonsei.ac.kr*
*Yonsei University*

**Sihun Baek**  *sihun.baek@duke.edu*
*Duke University*

**Jihong Park**  *jihong_park@sutd.edu.sg*
*Singapore University of Technology and Design*

**Hyelin Nam**  *hlnam@ramo.yonsei.ac.kr*
*Yonsei University*

**Praneeth Vepakomma**  *vepakom@mit.edu*
*MBZUAI and Massachusetts Institute of Technology*

**Ramesh Raskar**  *raskar@media.mit.edu*
*Massachusetts Institute of Technology*

**Mehdi Bennis**  *mehdi.bennis@oulu.fi*
*University of Oulu*

**Seong-Lyun Kim**[*]  *slkim@yonsei.ac.kr*
*Yonsei University*

**Reviewed on OpenReview:** *https://openreview.net/forum?id=4bo6XAnutd*

## Abstract

In computer vision, the vision transformer (ViT) has increasingly superseded the convolutional neural network (CNN) for improved accuracy and robustness. However, ViT's large model sizes and high sample complexity make it difficult to train on resource-constrained edge devices. Split learning (SL) emerges as a viable solution, leveraging server-side resources to train ViTs while utilizing private data from distributed devices. However, SL requires additional information exchange for weight updates between the device and the server, which can be exposed to various attacks on private training data. To mitigate the risk of data breaches in classification tasks, inspired from the CutMix regularization, we propose a novel privacy-preserving SL framework that injects Gaussian noise into smashed data and mixes randomly chosen patches of smashed data across clients, coined *DP-CutMixSL*. Our analysis demonstrates that DP-CutMixSL is a differentially private (DP) mechanism that strengthens privacy protection against membership inference attacks during forward propagation. Through simulations, we show that DP-CutMixSL improves privacy protection against membership inference attacks, reconstruction attacks, and label inference attacks, while also improving accuracy compared to DP-SL and DP-MixSL.

---

[*]Corresponding author

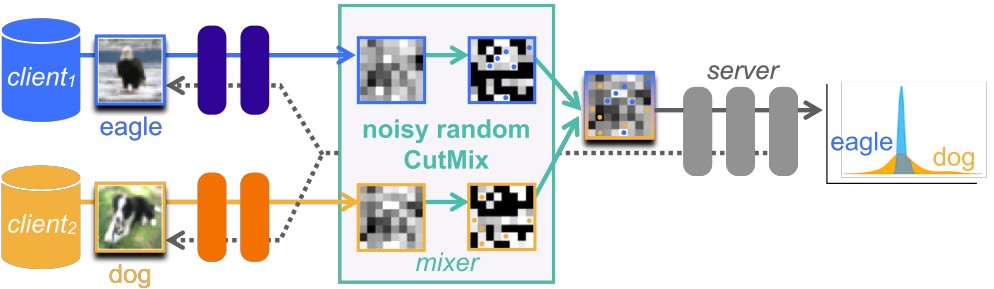

Figure 1: Schematic illustration of DP-CutMixSL with 2 clients.

# 1 Introduction

Transformer architecture has originally been developed in the domain of natural language processing (NLP) (Vaswani et al., 2017), and its application has recently been extended to various domains including speech recognition (Karita et al., 2019) and computer vision (CV) (Dosovitskiy et al., 2020b). In particular, the *vision transformer (ViT)* has recently been the new standard architecture in CV, succeeded to the convolutional neural network (CNN) architecture. ViT operations are summarized in two steps: 1) the first step to dividing image data into multiple image *patches*, and 2) the second step to learning the relationship between the patches under the encoder of the transformer. The latter step, dubbed the *self-attention* mechanism, helps to achieve high performance on large datasets, but causes performance degradation on small datasets. Hence, securing large-scale datasets in ViT is essential yet challenging, especially in distributed learning scenarios where huge data is dispersed to multiple *clients* with limited computing capability (Khan et al., 2021; Han et al., 2022). Federated learning (FL) is a promising solution in terms of enjoying these scattered data and computing resources (Li et al., 2020; Kairouz et al., 2021). In FL, each client trains a local model to be uploaded to the *server* with their own dataset, while the server yields the global model by taking the weighted average of the local models, leading to data diversity gain without direct data exchange. However, FL, which requires training the entire model on the client, is not suitable for ViT, which has a heavy computational burden and requires large memory due to its large model size.

To address this, *split learning (SL)* can be an alternative solution (Gupta and Raskar, 2018; Vepakomma et al., 2018). In this approach, an arbitrary single layer of the entire ViT model is defined as a *cut-layer*. The entire ViT model is then divided into a *lower model segment* and an *upper model segment* based on this cut-layer, with each segment stored on the client and server, respectively. During training in this model-split architecture, the client uploads the output from the cut-layer, referred to as *smashed data*, during forward propagation and downloads the corresponding gradient during back propagation. However, this exchange of information between the client and server can lead to privacy leakage. Unfortunately, the privacy leakage of ViT, to be shown in Figure 2a, is expected to be more severe than that of CNN, due to the absence of a pooling layer in ViT.

To this end, we propose a novel distributed learning framework for ViT, coined *DP-CutMixSL*, that is differentially private (DP) under the Parallel SL architecture. As depicted in Figure 1, each client of DP-CutMixSL uploads a portion of the smashed data according to the CutMix regularization (Yun et al., 2019), with additive Gaussian noise. Then, a novel entity, the *mixer*, combines these CutMixed noisy smashed data to generate the DP-CutMixSL's smashed data and propagates it to the server. By doing so, DP-CutMixSL gains in terms of robustness against various privacy attacks compared to uploading the smashed data itself. Specifically, we prove the effectiveness of DP-CutMixSL in protecting privacy against membership inference attack (Shokri et al., 2017; Rahman et al., 2018) and model inversion attack or reconstruction attack (He et al., 2019) in forward propagation, both theoretically and experimentally. In addition, we experimentally show the privacy guarantee for label inference attack (Li et al., 2021; Yang et al., 2022) in back propagation and demonstrate that high accuracy is also achieved thanks to the regularization effect of CutMix.

**Contributions.** The key contributions of this article are summarized as follows[1]:

- Inspired by CutMix, we propose a new SL-based architecture named DP-CutMixSL aiming to improve privacy guarantee of ViT. In this process, we introduce an entity called mixer on a network consisting of client-server, and outline its specific operations.

- We theoretically derive the privacy guarantee of DP-CutMixSL against membership inference attacks through DP analysis and experimentally demonstrate it. Through experiments, we verify that DP-CutMixSL is robust against reconstruction attack and label inference attack.

- In addition, we show that DP-CutMixSL outperforms baselines such as Parallel SL in terms of accuracy via numerical evaluation.

## 2 Related Works

**Vision Transformers.** The transformer architecture is first used in the NLP field (Vaswani et al., 2017), where its core operation is rooted on self-attention mechanism as well as encoder structure with multi-layer perceptron (MLP) and residual connection. In NLP, such transformer-based architecture is extended from Bidirectional Encoder Representations from Transformers (BERT) (Devlin et al., 2018), Generative Pre-trained Transformer (GPT) (Radford et al., 2018) to GPT-2 Radford et al. (2019), GPT-3 (Brown et al., 2020). This paradigm shift from CNN to transformer has reached out to the CV field. ViT, proposed in Dosovitskiy et al. (2020a), is the first of its kind to apply the transformer architecture to the CV field. ViT transforms an input image into a series of image patches, just as a transformer embeds words in text, and learns relationships between image patches, thereby a large-scale dataset is indispensable. This ViT operation enables to extract global spatial information, leading to its robustness against information loss such as patch drop and image shuffling compared to CNN (Naseer et al., 2021).

If most of the ViT works are based on the centralized method (Carion et al., 2020; Zheng et al., 2021; Chen et al., 2021), several studies have conducted research on distributed implementation of transformer or ViT (Hong et al., 2021; Park et al., 2021b; Qu et al., 2021). Hong et al. (2021) has designed FL-based transformer structure targeting text to speech task, while Qu et al. (2021) has explored the performance of FL in ViT when data are heterogeneous. To diagnose COVID-19, Park et al. (2021b) proposed SL-based architecture in ViT, benefiting from its robustness on task-agnostic training.

**Federated & Split Learning.** The key element of the distributed learning framework is to utilize raw data and computing resources spread across the sheer amount of Internet-of-Things (IoT) devices or clients. As the first kind of this, FL enables to acquire data diversity gain through exchanging model parameters (Li et al., 2020; Kairouz et al., 2021). FL's model parameter aggregation does not induce data privacy leakage, and what is more, it ensures scalability in terms of increasing accuracy with the number of participating clients (Konečný et al., 2015; Park et al., 2021a). Nevertheless, FL has a trouble in running a large-size model, constrained by its limited client-side computation and communication resources, highlighting the need for alternative solutions (Konečný et al., 2016; Singh et al., 2019). To this end, SL has appeared as a enabler for large model operation by splitting the entire model into two partitions (Gupta and Raskar, 2018; Vepakomma et al., 2018; Gao et al., 2020). The initial implementation of SL, which is based on sequential method, used to result in large latency especially with many clients, giving rise to the research on Parallel SL free from this problem. SFL, a combination of FL and SL, is the first form of Parallel SL, allowing simultaneous access by multiple clients (Thapa et al., 2020a;b; Gao et al., 2021). One step further, Pal et al. (2021), Oh et al. (2022b), and Oh et al. (2023) try to address the low accuracy, communication efficiency, and scalability of SFL.

**Privacy Attacks & Differential Privacy.** As machine learning develops rapidly, several types of privacy attacks have emerged whose goal is to extract information about training data, labels or the model

---

[1]This work is an extended version of both our previous workshop papers (Baek et al., 2022; Oh et al., 2022a), with the addition of extensive experiments involving label inference attacks and an analysis of the subsampled mechanism.

itself. In particular, regarding the privacy attack on distributed learning, Nasr et al. (2018) shows the membership inference attack of an adversary with some auxiliary information on the training data. He et al. (2019) investigates the reconstruction attack occurring on the inference phase of vanilla SL under white-box and black-box settings, while Oh et al. (2022b) measures it emprically on the Parallel SL structure. In addition, for label inference attacks in vanilla SL, Li et al. (2021) handles norm-based and direction-based attacks under black-box setting, and Yang et al. (2022) deals with white-box attacks and GradPerturb as a solution for them.

Accordingly, many studies have been conducted to protect information from various privacy attacks. One line of works first introduced the application of DP analysis technique to deep learning models (Dwork, 2008; Abadi et al., 2016). PixelDP, designed for SFL, is proposed as a DP-based defence to adversarial examples, providing certified robustness to AI/ML models, while Wu et al. (2022) applies the concept of DP to FL. Furthermore, Lowy and Razaviyayn (2021) defined record-level DP for practical use in cross-silo FL and inverstigated privacy protection between silos and the server. Meanwhile, Reńyi DP (RDP) is presented to facilitate the composition between heterogeneous mechanisms while providing tight bounds with fewer computations (Mironov, 2017).

Such differential privacy (DP) or Rényi differential privacy (RDP) bounds can be tightened via subsampling (Balle et al., 2018) and shuffling (Erlingsson et al., 2019). In particular, Yao et al. (2022) and Xu et al. (2024) provide experimental and theoretical studies on privacy-preserving split learning via patch shuffling on transformer structures. However, both studies are limited to single-client scenarios and do not address multi-client parallel computation or accuracy improvement in data-limited environments. Another method for privacy amplification is Mixup (Zhang et al., 2017; Verma et al., 2019), which leverages its inherent distortion property (Koda et al., 2021; Borgnia et al., 2021; Lee et al., 2019). In this context, our research exploits techniques for multi-user ViT by utilizing DP and CutMix, which inherently involve the concept of subsampling and present potential challenges when combined with shuffling.

## 3 Motivation: Privacy-Preserving Parallel SL in ViT

Consider a network with a set of clients $\mathbb{C} = \{1, 2, \cdots, n\}$ and a single server. Here, let $i$ be the subscript for the client. The dataset of the $i$-th client is consisting of multiple tuples of input data $\mathbf{x}_i$ and its one-hot encoded ground-truth label $\mathbf{y}_i$. We denote the $i$-th entire network as $\mathbf{w}_i = [\mathbf{w}_{c,i}, \mathbf{w}_s]^{\mathrm{T}}$, where $\mathbf{w}_{c,i}$, $\mathbf{w}_s$, and $(\cdot)^{\mathrm{T}}$ represent the $i$-th lower model segment, the upper model segment, and the transpose function, respectively.

Under the above setting, this section first revisits the Parallel SL operation on ViT. For the sake of convenience, we assume that the cut-layer is located between the embedding layer and the transformer in the ViT structure. This allows each $i$-th client to transform $\mathbf{x}_i$ into $\mathbf{s}_i$ consisting of multiple patches via $\mathbf{w}_{c,i}$. Then for all $i$, the $i$-th client sends $\mathbf{s}_i$, which is referred to as *smashed data*, to the server, followed by FP through $\mathbf{w}_s$ at the server, resulting in loss $L_i$. During the BP phase of Parallel SL, the server updates $\mathbf{w}_s$ based on aggregated losses $\sum_{i \in \mathbb{C}} L_i$ and sends *cut-layer gradients* to clients, who then update their respective $\mathbf{w}_{c,i}$.

By doing so, Parallel SL enables the client to efficiently offload the upper segment of the ViT to the server, which can be computationally expensive to run in its entirety, while still benefiting from exploring distributed data. However, Parallel SL with ViT has the following fundamental characteristics compared to it with CNN, as organized in Figure 2, which ultimately requires a solution considering these differences.

*1) Absence of Pooling Layer*: While CNNs typically include a pooling layer, ViTs often omit pooling layers, except in variants like the pooling-based ViT (PiT) (Heo et al., 2021b). Conversely, dropout layers in ViTs can impact the smashed data, whereas in CNNs, dropout layers usually affect the dense layers, leaving the earlier feature maps unchanged. When considering both pooling and dropout layers, the size of the smashed data changes significantly. For instance, using a $28 \times 28$ image, a $2 \times 2$ pooling layer with a stride of 2 reduces the data size to 25% of the original, whereas a dropout layer with a rate of 0.1 retains 90% of the data size without changing the spatial dimensions. This comparison highlights the difference in distortion levels between CNNs and ViTs, implying significant privacy leakage in ViTs. On the bright side, this makes regularization on hidden representations more fruitful, just like regularization on input data.

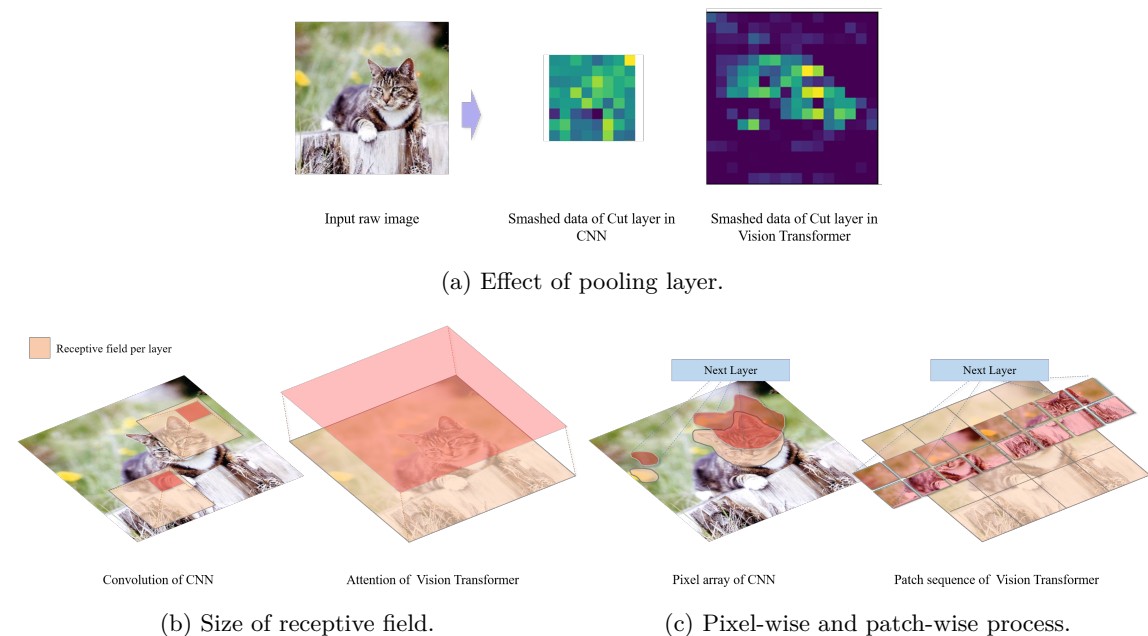

(a) Effect of pooling layer.

(b) Size of receptive field.

(c) Pixel-wise and patch-wise process.

Figure 2: Comparison of CNN and ViT operation from various perspectives.

*2) Large Receptive Field Size*: A CNN with a convolutional layer is specialized in catching local spatial information of an image, in other words, its receptive field size is small. Conversely, the receptive field of ViT is large enough to learn global spatial information, with the help of its self-attention mechanism. Because of this, ViT is more suitable for producing generalized models compared to CNN, but large-scale datasets are required to unleash the full potential of ViT due to its low inductive bias (Baxter, 2000). Data regularization can address this large-scale dataset requirement (Steiner et al., 2021).

*3) Patch-Wise Processing*: Due to the large receptive field of ViTs, as described in *2)*, they exhibit robustness against significant noise applied to parts of the image, such as patch drops or image shuffling (Naseer et al., 2021). Leveraging this inherent property, several studies (Yao et al., 2022; Xu et al., 2024) have attempted to design privacy-preserving frameworks for ViTs using patch-wise permutation and shuffling. Consequently, we intuitively infer that patch-wise operations may be more efficient for ViTs than pixel-wise operations in terms of both accuracy and privacy.

As highlighted in *1)*, smashed data in Parallel SL with ViT is vulnerable to privacy leakage. However, ViT's resilience to substantial noise in parts of the image, as observed in *2)*, paves the way for privacy-enhancing methods. In this context, CutMix regularization (Yun et al., 2019) emerges as a viable strategy where the client shares only a portion of their image while the accuracy can be guaranteed. The only thing to address is modifying CutMix to work patch-wise, as suggested in *3)*, which is covered in the next section. This patch-wise adaptation of CutMix, therefore, presents an integrated solution that aligns with the considerations from *1)* to *3)*, exploiting ViT's structural traits while maintaining data privacy.

## 4    Proposed: Split Learning With Random CutMix for ViT

In this section, we first propose a *Patch-Wise Random and Noisy CutMix (Random CutMix)*, aiming to improve ViT's privacy guarantee while still ensuring its accuracy. The key idea of Random Cutmix is that each client uploads a patch-wise fraction of the smashed data with additive Gaussian noise on top of the FP process in Parallel SL. Before getting into the details, we assume a network of client-mixer-server wherein the server is *honest-but-curious* with a privacy attack on data and labels and the mixer is a trusted third party. The mixer can be implemented through homomorphic encryption (Rivest et al., 1978; Pereteanu et al., 2022)

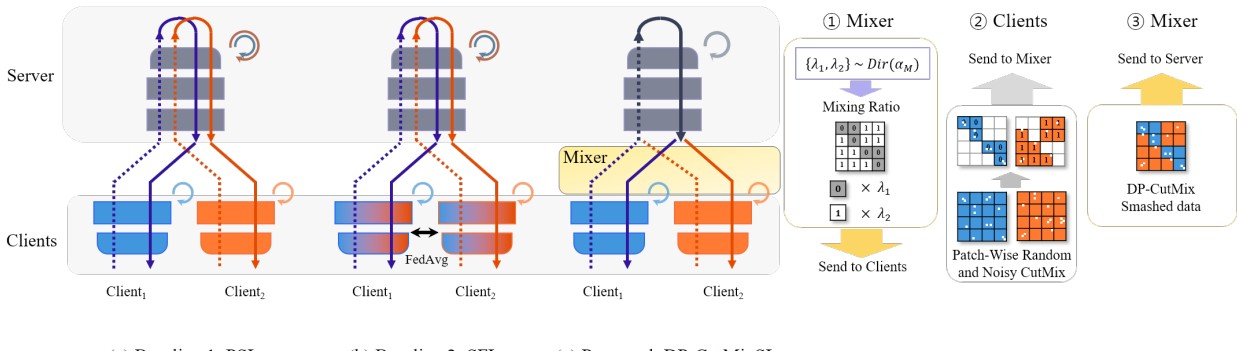

(a) Baseline 1: PSL.  (b) Baseline 2: SFL.  (c) Proposed: DP-CutMixSL.

Figure 3: Structural comparison of (c) DP-CutMixSL with (a) Parallel SL (PSL) and (b) split federated learning (SFL) (Thapa et al., 2020b). (1) In DP-CutMixSL, the mixer first calculates the $i$-th *mixing ratio* $\lambda_i$ following the symmetric dirichlet distribution (Bishop et al., 2007) with mask distribution $\alpha_M$. Depending on $\lambda_i$, the mixer creates the $i$-th mask $\mathbf{M}_i$, randomizing $\lceil \lambda_i \cdot N \rceil$ out of a total of $N$ patches. (2) The $i$-th client after the client-side FP punches the smashed data based on $\mathbf{M}_i$ and add Gaussian noise, which is then sent to the mixer. (3) The mixer consolidates the patch-wise randomly selected and noise-augmented smashed data received from clients, producing the smashed data of DP-CutMixSL with all patches intact. This is then transmitted to the server for the remaining SL operations including server-side forward propagation process.

that enables encrypted computation on the server side or analog communication as in Koda et al. (2021). Specifically, with homomorphic encryption, computation can be processed while preserving privacy. The client transmits homomorphically encrypted smashed data to the server, which then processes the data and produces encrypted output as a mixer.

Consider a network with a client number $n$ of 2. First, the mixer generates a patch-wise mask $\mathbf{M}_i$ with the given *mask distribution* $\alpha_M$ and sends it to the $i$-th client ($i \in \{1, 2\}$). After finishing the forward propagation process on the $i$-th lower model segment of SL with ViT, each client punches the smashed data $\mathbf{s}_i$ based on the mask and adds random noise with Gaussian distribution to it. The label of this is determined by the ratio $\lambda_i$ of the number of patches in the smashed data punched by $\mathbf{M}_i$ to the total number $N$ of patches. Accordingly, the $i$-th client sends the following $(\bar{\mathbf{s}}_i, \bar{\mathbf{y}}_i)$ pair to the server:

$$\bar{\mathbf{s}}_i = \mathbf{M}_i \odot (\mathbf{s}_i + n_{s,i}), \qquad \bar{\mathbf{y}}_i = \lambda_i \cdot (\mathbf{y}_i + n_{y,i}), \tag{1}$$

where $n_{s,i}$ and $n_{y,i}$ are matrices for the random noise added to the smashed data and label, whose elements follow a zero-mean Gaussian distribution with variances $\sigma_s^2$ and $\sigma_y^2$, respectively, and $\odot$ is an pixel-wise multiplication operator. Then, mixer aggregates $(\bar{\mathbf{s}}_i, \bar{\mathbf{y}}_i)$ to generate the output of Random CutMix ($\tilde{\mathbf{s}}_{\{1,2\}} = \bar{\mathbf{s}}_1 + \bar{\mathbf{s}}_2$, $\tilde{\mathbf{y}}_{\{1,2\}} = \bar{\mathbf{y}}_1 + \bar{\mathbf{y}}_2$) and sends it to server, yielding a loss $\tilde{L}_{\{1,2\}}$ via server-side FP.

In the back propagation phase, when the mixer receives cut-layer gradient $\nabla_{\tilde{\mathbf{s}}_{\{1,2\}}} \tilde{L}_{\{1,2\}}$ from the server, the mixer divides the gradient by client as shown in the following formula and sends it to each client:

$$\nabla_{\tilde{\mathbf{s}}_{\{1,2\}}} \tilde{L}_{\{1,2\}} = \mathbf{M}_1 \odot \nabla_{\tilde{\mathbf{s}}_{\{1,2\}}} \tilde{L}_{\{1,2\}} + \mathbf{M}_2 \odot \nabla_{\tilde{\mathbf{s}}_{\{1,2\}}} \tilde{L}_{\{1,2\}} \tag{2}$$

$$= \underbrace{\nabla_{\tilde{\mathbf{s}}_{\{1,2\}}} (\mathbf{M}_1 \odot \tilde{L}_{\{1,2\}})}_{\text{gradient for client 1}} + \underbrace{\nabla_{\tilde{\mathbf{s}}_{\{1,2\}}} (\mathbf{M}_2 \odot \tilde{L}_{\{1,2\}})}_{\text{gradient for client 2}}. \tag{3}$$

For a given gradient, each client and server updates the weights $\mathbf{w}_{c,i}, \mathbf{w}_s$ of the lower model segment and the upper model segment, completing a single round of *differential private SL with Random CutMix (DP-CutMixSL)*.

As shown in Equation 1, in DP-CutMixSL's forward propagation, each client's smashed data is randomly disclosed on a patch-by-patch basis with noise added to ensure the privacy guarantee for smashed data, and its back propagation similarly ensures the privacy guarantee for labels through the process shown in

Equation 3. Furthermore, when defining the number of clients performing Random CutMix, so-called *mixing group size* $k \leq n$, the aforementioned DP-CutMixSL can be considered as a case where $n = k = 2$, and observations on the performance of DP-CutMixSL for general $k$ can be found in Appendix A. Moreover, the detailed operation of Random CutMix and DP-CutMixSL is organized in Figure 3 and the pseudocode in Appendix C, respectively.

# 5 Differential Privacy Analysis on Smashed Data

In this section, we theoretically analyze the differential privacy (DP) bound of DP-CutMixSL and validate its effectiveness for privacy guarantees. Unlike existing DP works that focus on the privacy guarantees of samples and labels, we conduct DP analysis from the perspective of smashed data, which is highly correlated with the sample particularly in ViT, and its labels. This is in the same context as analyzing the privacy leakage of model parameters or gradient in FL. Note that in DP-CutMixSL, the component designed to safeguard against privacy breaches by leveraging a trusted intermediary known as the mixer, and the component that introduces Gaussian noise, both adhere to the Central and Gaussian Differential Privacy configurations, respectively. the definition of Central DP (CDP) (Dwork et al., 2006) is organized as follows:

**Definition 1** (($\varepsilon,\delta$)-CDP)**.** *For $\varepsilon \geq 0$ and $\delta > 0$, we say that a randomized mechanism $\mathcal{M} : \mathcal{D} \to \mathcal{R}$ is $(\varepsilon,\delta)$-CDP if it satisfies the following inequality for any adjacent $D, D' \in \mathcal{D}$ and $U \subset \mathcal{R}$:*

$$Pr[\mathcal{M}(D) \in U] \leq e^{\varepsilon} \cdot Pr[\mathcal{M}(D')] \in U] + \delta. \tag{4}$$

At this point, a small $\varepsilon$ indicates a high privacy level implying that one cannot distinguish whether $D$ or $D'$ is used to produce an outcome of mechanism. Although CDP is widely used when analyzing Gaussian mechanisms, we also use the Reńyi DP (RDP) (Mironov, 2017), defined below, given the tractable interpretation of its composition rule:

**Definition 2** (($\alpha, \epsilon$)-RDP)**.** *A randomized mechanism $\mathcal{M} : \mathcal{D} \to \mathcal{R}$ is said to have $\epsilon$-RDP of order $\alpha$, or $(\alpha, \epsilon)$-RDP for short, if for any adjacent $D, D' \in \mathcal{D}$ it holds that:*

$$D_{\alpha}(\mathcal{M}(D)\|\mathcal{M}(D')) \leq \epsilon. \tag{5}$$

In addition, Mironov (2017) proves that every RDP mechanism is also $(\varepsilon, \delta)$-CDP. Especially in Mironov (2017), when the mechanism is $(\alpha, \epsilon)$-RDP, then it is $(\epsilon + \frac{\log(1/\delta)}{\alpha - 1}, \delta)$-CDP for $0 < \delta < 1$.

We consider a scenario where each $i$-th client has one smashed data-label pair. Then, given the ViT's patch size of $P$, the full dataset on the client side consists of $n$ clients' pairs of smashed data $\mathbf{s}_i \in \mathbb{R}^{P^2 \times N \times C} = \mathbb{R}^{D_s}$ and the corresponding label $\mathbf{y}_i \in \mathbb{R}^L = \mathbb{R}^{D_y}$ is a one-hot vector of size $L$, where $N$ and $C$ denote the number of patches and channels, respectively ($\mathcal{D} = \{(\mathbf{s}_1, \mathbf{y}_1), .., (\mathbf{s}_n, \mathbf{y}_n)\}$). For ease of analysis, we assume that $\mathbf{s}_i$ and $\mathbf{y}_i$ are normalized so that $\mathbf{s}_i \in [0, \Delta]^{D_s}$ and $\mathbf{y}_i \in [0, 1]^{D_y}$, respectively. Based on the above premise, we first perform the RDP analysis on a single epoch.

Table 1: Comparison of mechanism output and RDP bounds for DP-CutMixSL, DP-MixSL and DP-SL.

| | Smashed Data | | Label | |
|---|---|---|---|---|
| | Output | RDP Bounds | Output | RDP Bounds |
| **DP-CutMixSL** | $\sum_{i=1}^{k} (\mathbf{M}_i \odot \mathbf{s}_i + n_{s,i})$ | $\lambda_{max} \cdot \epsilon_{1,s}(\alpha)$ | $\sum_{i=1}^{k} (\lambda_i \cdot \mathbf{y}_i + n_{y,i})$ | $\lambda_{max}^2 \cdot \epsilon_{1,y}(\alpha)$ |
| DP-MixSL | $\sum_{i=1}^{k} (\lambda_i \cdot \mathbf{s}_i + n_{s,i})$ | $\lambda_{max}^2 \cdot \epsilon_{1,s}(\alpha)$ | $\sum_{i=1}^{k} (\lambda_i \cdot \mathbf{y}_i + n_{y,i})$ | $\lambda_{max}^2 \cdot \epsilon_{1,y}(\alpha)$ |
| DP-SL | $\mathbf{s}_i + n_{s,i}$ | $\epsilon_{1,s}(\alpha)$ | $\mathbf{y}_i + n_{y,i}$ | $\epsilon_{1,y}(\alpha)$ |

As a comparison group for DP-CutMixSL, we use *DP-SL*, which applies the Gaussian mechanism to the existing SL, and *DP-MixSL*, which utilizes Mixup instead of CutMix in DP-CutMixSL. Table 1 shows the smashed data and labels of DP-CutMixSL compared to those of DP-MixSL or DP-SL. We also refer to the

differentially private mechanisms DP-SL, DP-MixSL, and DP-CutMixSL as $\mathcal{M}_1$, $\mathcal{M}_2$, and $\mathcal{M}_3$, respectively. Then, we can compare the RDP bound of DP-CutMixSL to those of DP-SL and DP-MixSL as follows:

**Theorem 1.** *For a given order $\alpha \geq 2$, the RDP privacy budgets $\epsilon_1(\alpha)$, $\epsilon_2(\alpha)$, and $\epsilon_3(\alpha)$ of DP-SL, DP-MixSL and DP-CutMixSL satisfy the following inequality if and only if its equalities hold when $\lambda_{max} = 1/n$:*

$$\epsilon_2(\alpha) \leq \epsilon_3(\alpha) \leq \epsilon_1(\alpha), \tag{6}$$

*where*

$$\epsilon_1(\alpha) = \epsilon_{1,s}(\alpha) + \epsilon_{1,y}(\alpha), \tag{7}$$

$$\epsilon_2(\alpha) = \lambda_{max}^2(\epsilon_{1,s}(\alpha) + \epsilon_{1,y}(\alpha)), \tag{8}$$

$$\epsilon_3(\alpha) = \lambda_{max}(\epsilon_{1,s}(\alpha) + \lambda_{max} \cdot \epsilon_{1,y}(\alpha)), \tag{9}$$

*in which $\epsilon_{1,s}(\alpha) = \frac{\alpha \Delta^2 D_s}{2\sigma_s^2}$, $\epsilon_{1,y}(\alpha) = \frac{\alpha D_y}{2\sigma_y^2}$, and $\lambda_{max} = \max_{i \in \mathbb{C}} \lambda_i$.*

Proof. *Combining Propositions 1 through 3 in Appendix E completes the proof.* ∎

Theorem 1 provides the following 4 observations about DP-CutMixSL.

**Privacy-Accuracy Trade-Offs ($k = n$).** There are two *privacy-accuracy trade-offs* identified in Theorem 1. First, DP-CutMixSL provides a lower RDP guarantee than DP-MixSL but achieves higher accuracy, as to be demonstrated in Section 6. The Mixup technique of DP-MixSL results in a superposition of images, where each pixel corresponds to multiple overlapping images. As the number of superimposed images, or equivalently the number of clients, increases, it becomes more challenging to identify each individual image by observing a single pixel. In contrast, the CutMix technique of DP-CutMixSL aggregates non-overlapping masked images. Due to the masking, this aggregation certainly leaks less privacy compared to the unmasked noisy images in DP-SL. Nevertheless, each pixel in this masked image aggregation corresponds to a specific image, making it easier to identify and thus achieving higher accuracy while leaking more privacy than the superimposed images in DP-MixSL. Another trade-off is associated with $\lambda_{max} \in [1/n, 1]$. When $\lambda_{max}$ reaches $1/n$ (i.e., $|\mathbb{C}|$ increases or $\alpha_M$ goes to $\infty$), the privacy guarantee is maximized and the equality constraint for Theorem 1's inequalities is satisfied, but the accuracy decreases as shown in Figure 10b and Figure 10c of Appendix A, leading to the privacy-accuracy trade-off.

**RDP-CDP Conversion.** Leveraging the compositional benefits of the RDP framework, the outcome of Theorem 1, initially derived from an analysis of a single epoch, can be straightforwardly generalized to scenarios involving multiple epochs. This extension is possible by simply applying the RDP sequential rule, but results in RDP bounds that worsen linearly with the number of epoch (Mironov, 2017; Feldman et al., 2018). Some recent work (Ye and Shokri, 2022; Altschuler and Talwar, 2022) attempts to derive tighter RDP bounds with multi epochs, but this is beyond our scope and we refer this as future work.

We can also measure the CDP guarantee of DP-CutMixSL by applying the aforementioned RDP-to-CDP conversion to Theorem 1, which is based on the RDP guarantee. Additionally, the effect of the mixing group size can be reflected by using the CDP bound formula of the subsampled mechanism in Wang et al. (2019), whereas Theorem 1 implicitly assumes that the mixing group size is equal to $n$. Thereby, for $k < n$, we can derive the CDP guarantee of DP-CutMixSL with subsampling ($\mathcal{M}^3 \circ$ **subsample**) as below, whose key operation consists of 1) a subsampling mechanism that randomly selects $k$ out of a total of $n$ datapoints (reflecting the mixing group size), and 2) operation of $\mathcal{M}^3$ as depicted in Figure 4.

**Corollary 1.** *For all integer $\alpha \geq 2$ and $0 < \delta < 1$, the DP privacy budgets $\varepsilon_1'(\delta)$, $\varepsilon_2'(\delta)$, and $\varepsilon_3'(\delta)$ of $\mathcal{M}^1 \circ$ **subsample**, $\mathcal{M}^2 \circ$ **subsample** and $\mathcal{M}^3 \circ$ **subsample** satisfy the following inequality:*

$$\varepsilon_2'(\delta) \leq \varepsilon_3'(\delta) \leq \varepsilon_1'(\delta), \tag{10}$$

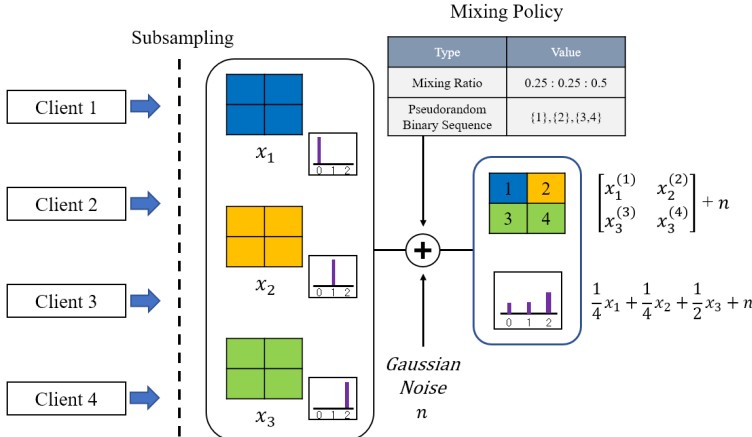

Figure 4: An illustration of DP-CutMixSL with subsampling when $n = 4$ and $k = 3$.

*where*

$$\varepsilon'_1(\delta) = \log\left(1 + \frac{k}{n}(e^{\epsilon_1(\alpha) + \varepsilon_o(\delta)} - 1)\right), \tag{11}$$

$$\varepsilon'_2(\delta) = \log\left(1 + \frac{k}{n}(e^{\epsilon_2(\alpha) + \varepsilon_o(\delta)} - 1)\right), \tag{12}$$

$$\varepsilon'_3(\delta) = \log\left(1 + \frac{k}{n}(e^{\epsilon_3(\alpha) + \varepsilon_o(\delta)} - 1)\right), \tag{13}$$

*in which $k_2^* = \sqrt{\frac{\epsilon_{1,s}(\alpha) + \epsilon_{1,y}(\alpha)}{\varepsilon_o(\delta)}}$ and $k_3^* = \sqrt{\frac{\epsilon_{1,y}(\alpha)}{\varepsilon_o(\delta)}}$ minimize $\varepsilon'_2(\delta)$ and $\varepsilon'_3(\delta)$ under the assumption that $\lambda_i = 1/k \;\forall i$, respectively, and $\varepsilon_o(\delta) = \frac{\log(1/\delta)}{\alpha - 1}$.*

Sketch of Proof. *Recall Theorem 1, and apply it to the DP bound formula of the subsampled mechanism of Wang et al. (2019) (if $\mathcal{M}$ is $(\varepsilon, \delta)$-DP, then the subsampled mechanism $\mathcal{M} \circ \textbf{subsample}$ is $(log(1 + \gamma(e^{\varepsilon} - 1)), \gamma\delta)$-DP where $\gamma$ denotes sampling ratio). This yields Equation 10.*

*Assuming $\max_{i \in \mathbb{C}} \lambda_i = 1/k$, for $\epsilon_3(\alpha) + \varepsilon_o(\delta) \ll 1$, $\varepsilon'_3(\delta) = \log\left(1 + \frac{k}{n}(e^{\epsilon_3(\alpha) + \varepsilon_o(\delta)} - 1)\right)$ is approximated by $\log\left(1 + \frac{k}{n}(\epsilon_3(\alpha) + \varepsilon_o(\delta))\right)$. Since the log function is a monotone increasing function and $n$ is fixed, $k \cdot (\epsilon_3(\alpha) + \varepsilon_o(\delta))$ should be minimized for the minimum $\varepsilon'_3(\delta)$. Regarding $k \cdot (\epsilon_3(\alpha) + \varepsilon_o(\delta))$, since it is a convex function for $k > 0$, we can find $k_3^*$ which becomes 0 when differentiated. $k_2^*$ can also be calculated in a similar manner. This completes the proof.* ∎

**Revisiting Privacy-Accuracy Trade-Off ($k < n$).** First, privacy-accuracy trade-off between DP-MixSL and DP-CutMixSL occurs in Corollary 1 as in Theorem 1 where $n = k$. The existence of an optimal mixing group size is rooted in subsampling. In the existing Mixup or Random CutMix, the privacy guarantee improves when the number of samples increases, but counterintuitively, with subsampling, the randomness of which client among all clients a specific sample belongs to decreases, resulting in a trade-off.

**Limitations on DP Analysis.** There are 2 major limitations on our DP analysis. Firstly, in existing DP analysis, it fundamentally measures how sensitively the output changes compared to the input, and at this time, the output is in-practice bounded (for example, classification). However, since SL inherently lacks in quantifying the change of smashed data versus input, we indirectly analyze the privacy guarantee of smashed data versus output. To complement this, we experimentally measure the attack success rate for membership inference attacks and assess robustness against reconstruction attacks in Section 6 to ensure privacy guarantees between input and smashed data.

Secondly, DP analysis cannot differentiate between Random CutMix and Vanilla CutMix because it focuses on the quantity rather than the randomness or pattern of the mechanism. Through the robustness of the

reconstruction attack in Section 6 mentioned above, we bypass the privacy guarantee between CutMix, which is theoretically indistinguishable.

# 6 Numerical Evaluation

While Section 5 theoretically demonstrates the privacy guarantee of DP-CutMixSL, this section experimentally analyzes its privacy guarantee and accuracy compared to Parallel SL, SFL (Thapa et al., 2020b), and etc. For the experiment, we use the CIFAR-10 dataset (Krizhevsky, 2009) with a batch size of 128, and Table 4 additionally uses the Fashion-MNIST dataset (Xiao et al., 2017). As an optimizer, Adam with decoupled weight decay (AdamW) (Loshchilov and Hutter, 2017) is used with a learning rate of 0.001, and a total of 10 clients each have 5,000 images. Except for Table 4, we use the ViT-tiny model (Touvron et al., 2020), and the entire model is split so that the client and server each have an embedding layer and a transformer. Regarding the injection of noise into the one-hot encoded label, we use a clamp function to bound the values between 0 and 1 after the noise is added. Additionally, in Figure 5, for the case of $n = 10$ and $k > 2$, the mixer clusters a set of $n$ clients into subsets of size $k$ at each epoch (constructing each subset from the remaining clients) before determining the mixing ratio and masks within each subset. Furthermore, for the purpose of training the model, we employ an environment consisting of 64 patches, each measuring $2 \times 2$, and conduct training over a span of 600 epochs. Other parameters for DP measurement are in Table 2.

Table 2: Parameters for DP measurement.

| Parameter | Annotation | Value |
|---|---|---|
| Number of clients | $n$ | 10 |
| Dimension of smashed data | $D_s$ | 10 |
| Dimension of label | $D_y = L$ | 2 |
| Pixel-wise upper bound of smashed data | $\Delta$ | 0.15 |
| Mixing ratio | $\lambda_i \; \forall i$ | $1/k$ (uniform) |
| RDP parameter | $\alpha$ | 2 |
| DP parameter | $\delta$ | 0.0002 |

To measure the robustness of DP-CutMixSL against privacy attacks, we first consider the following three types of privacy attacks: membership inference attack, reconstruction attack, and label inference attack. Among them, membership inference attack and reconstruction attack are privacy attacks that occur in the forward propagation process of DP-CutMixSL and label inference attack in its back propagation process, respectively. Specifically, an honest but curious server in a membership inference attack attempts to determine whether a particular client's data is used for training, via the uploaded DP-CutMix's smashed data and label generated by the mixer in Equation 1. Similarly, in a reconstruction attack, the server aims to restore the input data of the client through the auxiliary network with the uploaded smashed data of DP-CutMixSL.

Furthermore, in the label inference attack, a *honest-but-curious* client tries to infer the label of input data used by another client through the cut-layer gradient. For example, assuming that the cut-layer gradient $\nabla_{\tilde{\mathbf{s}}_{\{1,2\}}} \tilde{L}_{\{1,2\}}$ in Equation 3 is sent to clients 1 and 2, the 2-th client can try to infer the label of the 1-th client by performing a classification with the 1-th client's gradient $\nabla_{\tilde{\mathbf{s}}_{\{1,2\}}} (\mathbf{M}_1 \odot \tilde{L}_{\{1,2\}})$ as input, which is included in the entire cut-layer gradient.

**Privacy Against Membership Inference Attacks.** Here we assume the worst case that the server knows the smashed data of all client. Given this premise, the DP analysis in Section 5 enables a theoretical evaluation of privacy protection against membership inference attacks, thus leading to the measurement of the CDP bound.

Figure 5a shows the accuracy of each method as a function of the CDP guarantee ($\varepsilon$). For the same $\varepsilon$, DP-CutMixSL achieves higher accuracy than both DP-SL and DP-MixSL, except when $\varepsilon = 1$. Appendix B presents the noise variance required for each method to guarantee privacy for each $\varepsilon$ in Figure 5a. Here, the noise variance increases in the order of DP-SL, DP-CutMixSL, and DP-MixSL, implying that the accuracy

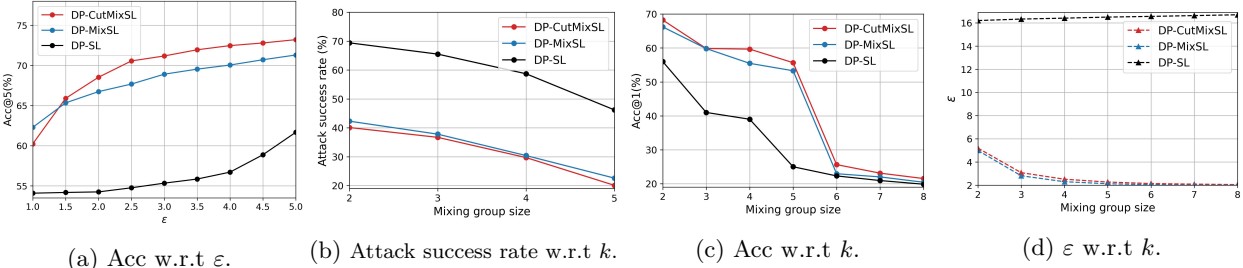

(a) Acc w.r.t $\varepsilon$.      (b) Attack success rate w.r.t $k$.      (c) Acc w.r.t $k$.      (d) $\varepsilon$ w.r.t $k$.

Figure 5: Accuracy, attack success rate, and $\varepsilon$ under the CIFAR-10 dataset: (a) accuracy of DP-CutMixSL, DP-MixSL, and DP-SL according to $\varepsilon$; (b) attack success rate of membership inference attacks against DP-CutMixSL, DP-MixSL, and DP-SL according to $k$; (c) accuracy of DP-CutMixSL, DP-SL, and DP-MixSL according to $k$; (d) $\varepsilon$ of DP-CutMixSL, DP-SL, and DP-MixSL according to $k$.

drop at $\varepsilon = 1$ in Figure 5a is attributed to the injection of large-scale noise. In fact, Figure 11a in Appendix D demonstrates that DP-CutMixSL outperforms DP-SL and DP-MixSL in terms of accuracy for all given noise variances. Additionally, Figure 11b illustrates the superiority of DP-CutMixSL as a privacy amplifier with an improved $\varepsilon$ compared to DP-SL.

Figure 5b, Figure 5c, and Figure 5d compare the impact of mixing group size on attack success rate, accuracy, and $\varepsilon$ for membership inference attacks, respectively. In particular, Figure 5b is measured using a neural network-based attacker model with three fully connected layers pretrained on the CIFAR-10 dataset. Both Figure 5b and Figure 5d show that DP-CutMixSL provides improved privacy guarantees against membership inference attacks compared to DP-SL. Factors that improve the DP guarantee of DP-CutMixSL include 1) noise injected by Gaussian mechanism and 2) DP guarantee amplification through Random CutMix, and such gap in privacy guarantee between DP-SL and DP-CutMixSL indicates that the latter is more critical to privacy. Comparing DP-CutMixSL and DP-MixSL, Figure 5b demonstrates the superiority of DP-CutMixSL, while Figure 5d shows the opposite. This highlights the aforementioned limitations of DP analysis and the experimental superiority of DP-CutMixSL.

In Figure 5c, DP-CutMixSL achieves higher accuracy compared to both DP-MixSL and DP-SL, regardless of the mixing group size. In Figure 5c and Figure 5d, both accuracy and $\varepsilon$ of DP-CutMixSL and DP-MixSL decrease as the mixing group size increases, while those of DP-SL tend to be reversed. This is because in the trade-off of privacy guarantee between subsampling and CutMix or Mixup, the privacy guarantee gain of CutMix or Mixup as $k$ increased is greater than the loss of privacy guarantee due to subsampling, resulting in a "Hiding in the crowd" effect (Jeong et al., 2020). It can also be explained by how large the optimal mixing group size is (convex function with respect to $k$), and large $k_2^*$ as well as $k_3^*$ for given parameters validate it ($k_2^* = 28.55$, $k_3^* = 27.07$). On the other hand, DP-SL lacks CutMix or Mixup, so a small $k$ leads to a strong privacy guarantee due to subsampling. Moving forward, to more effectively utilize CutMix's privacy protection capabilities without introducing noise, we assess performance in noiseless environments. Consequently, we are updating our naming conventions: DP-CutMixSL will now be referred to as CutMixSL, and DP-MixSL will be known as PSL with Mixup, among others.

**Privacy Against Reconstruction Attacks.** For the reconstruction attack, an auxiliary network is utilized, which takes the smashed data as input and produces *restored data* through two convolutional layers followed by interpolation. The auxiliary network is trained using the CIFAR-10 dataset by minimizing the mean-squared-error (MSE) loss between the restored data and the original input data. Regarding the hyperparameters, we adjust the dataset size for training the auxiliary network, mask distribution, and mixing group size.

Table 3 shows the reconstruction loss of SL-based methods according to various hyperparameters. When comparing SL-based techniques, the reconstruction loss of the proposed CutMixSL is the largest, in other words, CutMixSL outperforms in terms of privacy guarantee for reconstruction attack in most cases, followed by PSL w. Mixup. In particular, when comparing CutMixSL (Random CutMix) and PSL w. vanilla CutMix,

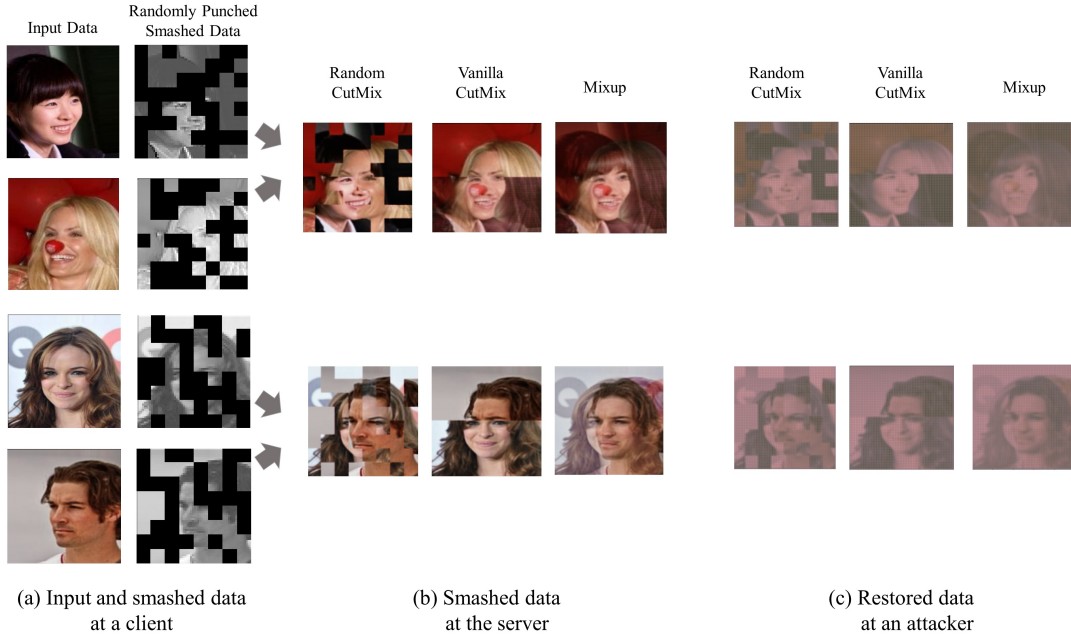

(a) Input and smashed data
at a client

(b) Smashed data
at the server

(c) Restored data
at an attacker

Figure 6: Example images of raw, smashed, and restored data for Random CutMix, Vanilla CutMix, and Mixup.

Table 3: Reconstruction loss (MSE) of SL-based techniques according to mixing group size, train dataset size, and mask distribution.

| | Training Dataset (10%) | | | | Training Dataset (100%) | | | |
|---|---|---|---|---|---|---|---|---|
| Mask Distribution ($\alpha_M$) | 2 | | 6 | | 2 | | 6 | |
| Mixing group size ($k$) | 2 | 4 | 2 | 4 | 2 | 4 | 2 | 4 |
| PSL | 0.403 | 0.425 | 0.326 | 1.425 | 0.116 | 0.308 | 0.138 | 0.398 |
| PSL w. Mixup | **0.665** | 0.383 | 0.379 | **1.923** | 0.172 | 0.396 | 0.215 | 0.292 |
| PSL w. Vanilla CutMix | 0.382 | 0.426 | 0.429 | 1.316 | 0.180 | **0.403** | 0.219 | 0.417 |
| CutMixSL (proposed) | 0.425 | **0.466** | **0.441** | 1.561 | **0.187** | 0.312 | **0.221** | **0.435** |

the robustness of CutMixSL is superior for most hyperparameter settings. This is due to the difference in randomness between Vanilla and Random CutMix, which is previously indistinguishable by DP analysis. Since adjacent box-shaped pixels are replaced, Vanilla CutMix has a relatively high correlation between pixels, while the correlation between pixels in a Random CutMix that is randomly replaced patch-wise is bounded in patch units, straightforwardly leading to a strong privacy guarantee.

With respect to the overall tendency for hyperparameters, the larger the training dataset size, the smaller the mask distribution, and the smaller the mixing group size, the more advantageous the auxiliary network is to learn the restored data, leading to a small reconstruction loss. Finally, Figure 6 presents examples from the CelebA dataset, showcasing the original data, the smashed data, and the corresponding restored data generated by the auxiliary network. This comparison highlights the superiority of the Random CutMix technique.

**Privacy Against Label Inference Attacks.** For label inference attacks, there are white-box attacks in Yang et al. (2022), black-box attacks in Li et al. (2021), and other minor variations. We consider a white-box attack among them, since black-box attacks include the bold assumption that clients know the upper model segment weight of the server. Unlike Li et al. (2021), which is based on Vanilla SL, we consider

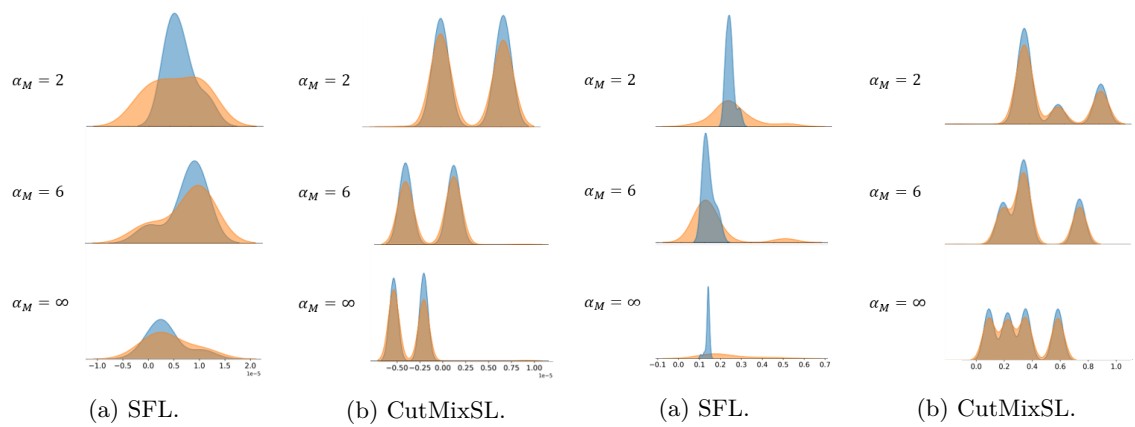

Figure 7: Norm distribution of gradients according to mask distribution.

Figure 8: Cosine similarity distribution of gradients according to mask distribution.

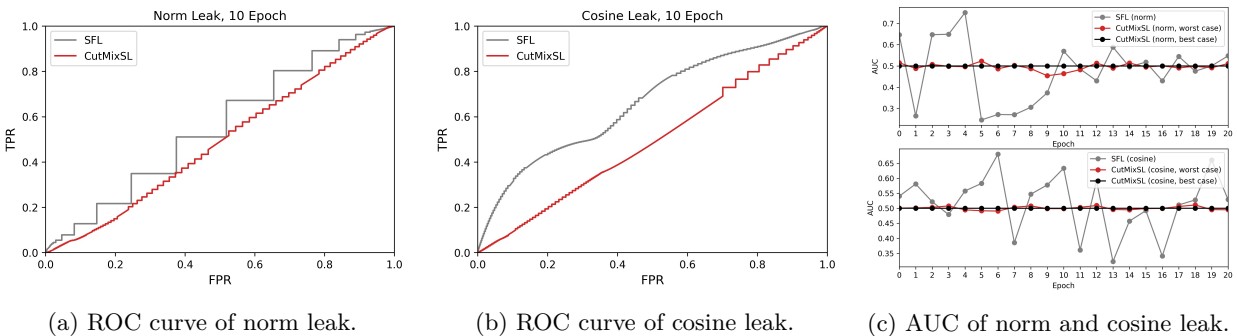

(a) ROC curve of norm leak.

(b) ROC curve of cosine leak.

(c) AUC of norm and cosine leak.

Figure 9: Privacy guarantee measurement for label inference attack of CutMixSL and SFL.

the following worst case of CutMixSL to enable powerful white-box attacks in parallelized form of SL: 1) the first assumption of weight averaging as in SFL to share the weight of the lower model segment among clients, 2) the second assumption that the gradient in the cut-layer is averaged as in Pal et al. (2021) before broadcasting to clients, 3) the third assumption that label inference leakage occurs between clients within the same mixing group size for CutMixSL with $k = 2$. Also, we refer to CutMixSL as its best case when not with the above assumptions. Then, a honest-but-curious client aims to infer its label by measuring the norm (*norm leak*) or cosine similarity (*cosine leak*) of the averaged cut-layer gradient as well as the gradient propagate to the lower model segment.

For measurement, we compute the *area under the ROC curve (AUC)* over the distribution of the norm or cosine similarity. The ROC curve means the curve of the true positive rate (TPR) and false positive rate (FPR) as the decision boundary moves from $-\infty$ to $\infty$ in a binary classification scenario, and if its base area, AUC, is close to 1, it means that the classification of the two distributions becomes clear under accurate data labeling. Thus, in our scenario, an AUC close to 0.5 implies a high privacy guarantee against label inference attacks. As an experimental setting, we utilize the LeNet-5 model (LeCun et al., 2015), where the cut-layer is located after the second convolutional layer, and allocate 1,000 samples each corresponding to the two labels 0 and 4 of MNIST dataset to two clients. As a comparator, we use SFL with cut-layer gradient averaging.

Figure 7 and 8 visualize the distribution of norm and cosine similarity according to the mask distribution of CutMixSL and SFL, respectively. where the orange and blue regions each represent that the labels are positive and negative. We can visually confirm that, as the mask distribution increases, CutMixSL does not change significantly, whereas in SFL, the variance of the distribution increases, making it easier to distinguish. Based on these, the ROC curves for norm leak and cosine leak are shown in Figure 9a and 9b.

Table 4: Top-1 accuracy of methods for various datasets and models.

| Method | Models w/ CIFAR-10 | | | Models w/ Fashion-MNIST | | |
|---|---|---|---|---|---|---|
| | ViT-Tiny | PiT-Tiny | VGG-16 | ViT-Tiny | PiT-Tiny | VGG-16 |
| Standalone | 48.84 | 47.77 | 54.97 | 77.65 | 78.21 | 80.12 |
| PSL | 57.21 | 52.28 | 62.62 | 85.68 | 82.35 | 84.39 |
| SFL | 67.88 | 55.63 | 63.98 | 89.17 | 84.27 | 87.34 |
| PSL w. Mixup | 69.23 | 64.89 | **68.20** | 88.21 | 87.62 | 88.53 |
| PSL w. Random Cutout | 53.86 | 50.28 | 56.65 | 88.46 | 86.48 | 88.17 |
| PSL w. Vanilla CutMix | 71.78 | 58.21 | 33.50 | 87.86 | 86.31 | 89.01 |
| CutMixSL (proposed) | **73.77** | **71.26** | 67.53 | **89.75** | **89.25** | **89.45** |

Further, Figure 9c measures AUC per epoch of SFL and CutMixSL (its worst case as well as its best case) for norm and cosine leaks. The first thing to note is the strong privacy guarantee for norm and cosine leak of CutMixSL for both cases, maintaining an AUC close to 0.5 for all epochs, thanks to parallelization and Random CutMix's masking effect on gradient of Equation 3. The baseline SFL reaches AUCs up to 0.75 and 0.68 for norm and cosine leak, respectively, roughly alleviated by parallelization alone. In addition, regarding the impact of norm and cosine leak according to epoch, the variance of norm leak AUC is larger at the beginning of learning, but becomes weaker as epoch progresses, and instead, the variance of cosine leak AUC becomes large, showing the potential complementary threats of the two privacy attacks.

**Accuracy under IID Dataset.** In Table 4, we additionally measure the accuracy for PiT-tiny Heo et al. (2021a) and VGG-16 Simonyan and Zisserman (2014) except for ViT-tiny. Here, VGG-16 is for CNN in addition to ViT, and PiT is a model between ViT and CNN and is a transformer architecture equipped with a pooling layer. For an extensive comparison of results, we consider PSL with Random Cutout, which is equivalent to the single client case of Random CutMix.

Table 4 shows the top-1 accuracy on the CIFAR-10 and Fashion-MNIST datasets of various SL-based techniques, including CutMixSL. First, the accuracy of CutMixSL is the highest in all cases except for the case where VGG-16 and CIFAR-10 are used. With VGG-16 and CIFAR-10, PSL w. Mixup achieves the highest accuracy. This is because, as mentioned earlier, CNN focuses on locality when learning spatial information, while ViT focuses on globality. Also, it is consistent with Naseer et al. (2021), indicating that ViT has robustness of accuracy against patch drop or image shuffling compared to CNN. For that reason, CNN and ViT are better suited for superposition type regularization (i.e., Mixup) and masking type regularization (i.e., Cutout, CutMix), respectively Harris et al. (2020). Compared to PSL w. Vanilla CutMix, CutMixSL demonstrates superior accuracy, validating our intuition about the efficacy of the patch-wise designed regularizer in ViT. From the perspective of dropout Srivastava et al. (2014), it can be also seen that the increased randomness in CutMixSL results in higher accuracy. Furthermore, as in Appendix F, Random CutMix achieves the highest accuracy even when applied at the input layer. Straightforwardly, Random CutMix applied to the input layer, however, is more vulnerable to data privacy leakage than that applied to the cut-layer, resulting in an *accuracy-privacy trade-off*.

## 7 Conclusion

In this study, we designed DP-CutMixSL with the goal of developing a privacy preserving distributed ML algorithm for ViT. Thanks to the randomness and masking effect of Random CutMix, we theoretically and experimentally demonstrated that the proposed DP-CutMixSL has robustness against three types of privacy attacks, while not compromising accuracy. While this work focuses on an SL-based algorithm that enables privacy-preserving and accurate parallel computation in multi-user ViTs, it also shows promise in relation to existing techniques for privacy-preserving SL in single-user ViTs. Notably, we briefly examined the scalability and privacy-accuracy trade-offs of patch-wise shuffling (Yao et al., 2022; Xu et al., 2024) and its application to DP-CutMixSL in Tables 7 and 8 of Appendix G, which are worthy of further exploration in future research.

Although DP guarantee for smashed data was theoretically derived in FP, but our study lacks it in BP. Combined with GradPerturb in Yang et al. (2022), exploring the DP guarantee at BP could be an interesting topic for future work. Furthermore, while controlling the mixing group size only in this work, it is possible to increase the number of FP flows by combinatorily setting the mixing group several times during single FP as in Oh et al. (2022b), focusing on its augmentation properties. This can be a solution to the low inductive bias of ViT, which is deferred to future research.

### Broader Impact Statement

In this paper, we introduce a novel parallel training method for transformer structures designed to enhance privacy guarantees while improving accuracy for multi-client environments, leveraging memory-efficient structures based on split learning. We believe our work can significantly contribute to privacy-preserving training schemes for memory-constrained devices, offering robust protection against membership inference attacks, model inversion attacks, and label inference attacks. However, real-world implementation can be challenging, particularly regarding the implementation of a mixer. Although homomorphic encryption and AirComp can be adopted without additional deployment costs, factors such as encryption/decryption speed especially for large deep learning models and time synchronization must be carefully considered for each. Therefore, we recommend a rigorous approach to implementing our framework in real-world scenarios, taking into account deployment costs, latency requirements, and other relevant factors.

### Acknowledgments

This work was supported in part by the Korean Ministry of Science and ICT (MSIT) under the National Research Foundation of Korea (NRF) grant (No. 2023-11-1836) and the Information Technology Research Center (ITRC) support program (IITP-2024-RS-2023-00259991) supervised by IITP.

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

# Appendices

## A Observations on Random CutMix

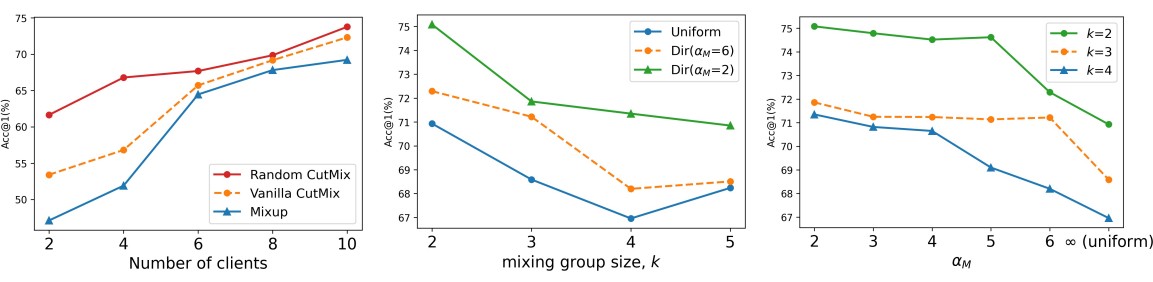

(a) Impact of mixing methods.  (b) Impact of mixing group size.  (c) Impact of mask distributions ($\alpha_M$).

Figure 10: Top-1 accuracy of multi-client scenario: (a) accuracy of Random CutMix, Vanilla CutMix, and Mixup w.r.t the number of clients; (b) accuracy of Random CutMix w.r.t the mixing group size; (c) accuracy of Random CutMix w.r.t the mask distribution $\alpha_M$.

In this subsection, the design elements of Random CutMix are explored, along with its optimal hyperparameter settings, especially with respect to its accuracy.

**Random CutMix vs. Vanilla CutMix and Mixup.** While the proposed Random CutMix is a patch-wise partial regularization scheme rooted in masking, a Mixup (Zhang et al., 2017; Verma et al., 2019) that superpositions the full image can be considered as an alternative. Another alternative is Vanilla CutMix, a masking type regularization equal to Random CutMix. Figure 10a shows the comparison of accuracy between these regularization schemes according to the number of clients $n$. The top-1 accuracy is high in the order of Random CutMix, Vanilla CutMix, and Mixup regardless of the number of clients, showing superiority of Random CutMix in ViT. Also, in all three regularization schemes, accuracy increases as the number of clients increases, that is, scalability is guaranteed up to 10 clients.

**Impacts of Mixing Group Size and Mask Distributions.** In Figure 10b, the accuracy of the Random CutMix as the mixing group size varies is shown for mask distribution $\alpha_M$. Note here that the infinite divergence of $\alpha_M$ implies that the mixing ratio follows a uniform distribution. Without cases where $k$ is 4 with $\alpha_M$ of 2 or 6, the top-1 accuracy tends to be inversely proportional to the mixing group size, especially its decline is greatest when $k$ changes from 2 to 3. Interpreting this from the perspective of each client, as the mixing group size $k$ increases, large noise that may lead to performance degradation is applied in the remaining areas except for $\frac{1}{k}$ of the entire image, under the assumption of a uniform mixing ratio. For similar reasons regarding distortion level, Figure 10c includes a tendency for accuracy to decrease as $\alpha_M$ increases.

## B Noise Variance Settings in Figure 5a

| | Noise variance ($\sigma_s^2$, $\sigma_y^2$) | | | | | | | | |
|---|---|---|---|---|---|---|---|---|---|
| $\varepsilon$ | 1 | 1.5 | 2 | 2.5 | 3 | 3.5 | 4 | 4.5 | 5 |
| DP-SL | 246/255 | 164/255 | 123/255 | 98.5/255 | 80/255 | 70.5/255 | 61/255 | 55/255 | 49/255 |
| DP-CutMixSL | 66/255 | 47/255 | 36/255 | 29/255 | 25/255 | 21/255 | 19/255 | 17/255 | 15/255 |
| DP-MixSL | 60/255 | 43/255 | 33/255 | 27/255 | 23/255 | 20/255 | 17/255 | 15/255 | 14/255 |

Table 5: Noise variances for DP-SL, DP-MixSL, and DP-CutMixSL for various values of $\varepsilon$.

## C  DP-CutMixSL's Pseudocode

---

**Algorithm 1** Operation of DP-CutMixSL with $n = k = 2$.

---

**Input:** mask distribution $\alpha_M$

  /\*Execute in mixer\*/

  **function 1.** Pseudorandom Sequence Generation

    Sample $\{\lambda_1, \lambda_2\} \sim \text{Dir}(\alpha_M)$

    Generate binary masks $\{\mathbf{M}_1, \mathbf{M}_2\}$ according to $\{\lambda_1, \lambda_2\}$

    **Return** $\mathbf{M}_i$ to $i$-th client for all $i \in \{1, 2\}$

  **function 2.** Random CutMix

    Generate $\{\tilde{\mathbf{s}}_{\{1,2\}}, \tilde{\mathbf{y}}_{\{1,2\}}\}$ through $\tilde{\mathbf{s}}_{\{1,2\}} = \bar{\mathbf{s}}_1 + \bar{\mathbf{s}}_2$, $\tilde{\mathbf{y}}_{\{1,2\}} = \bar{\mathbf{y}}_1 + \bar{\mathbf{y}}_2$

    **Return** $\{\tilde{\mathbf{s}}_{\{1,2\}}, \tilde{\mathbf{y}}_{\{1,2\}}\}$ to server

  **function 3.** Cut-layer Gradient Splitting

    Split $\nabla_{\tilde{\mathbf{s}}_{\{1,2\}}} \tilde{L}_{\{1,2\}}$ into $\nabla_{\tilde{\mathbf{s}}_{\{1,2\}}} \mathbf{M}_1 \odot \tilde{L}_{\{1,2\}}$ and $\nabla_{\tilde{\mathbf{s}}_{\{1,2\}}} \mathbf{M}_2 \odot \tilde{L}_{\{1,2\}}$ through Equation 3

    **Return** $\nabla_{\tilde{\mathbf{s}}_{\{1,2\}}} \mathbf{M}_i \odot \tilde{L}_{\{1,2\}}$ to $i$-th client

  **while** $\mathbf{w}_i$ not converged **do**

    Pseudorandom Sequence Generation

    /\*Execute in client $i$\*/

    Generate $\mathbf{s}_i$ by passing $\mathbf{x}_i$ through $\mathbf{w}_{c,i}$

    Generate $(\bar{\mathbf{s}}_i, \bar{\mathbf{y}}_i)$ as in Equation 1 based on $\mathbf{M}_i$ and Gaussian mechanism

    Send $(\bar{\mathbf{s}}_i, \bar{\mathbf{y}}_i)$ to mixer

    Random CutMix

    /\*Execute in server\*/

    Generate loss $\tilde{L}_{\{1,2\}}$ through server-side FP via $\mathbf{w}_s$

    Send $\nabla_{\tilde{\mathbf{s}}_{\{1,2\}}} \tilde{L}_{\{1,2\}}$ to mixer & Update $\mathbf{w}_s$

    Cut-layer Gradient Splitting

    /\*Execute in client $i$\*/

    Update $\mathbf{w}_{c,i}$

  **end while**

---

## D  Performance of DP-CutMixSL on Noise Variance

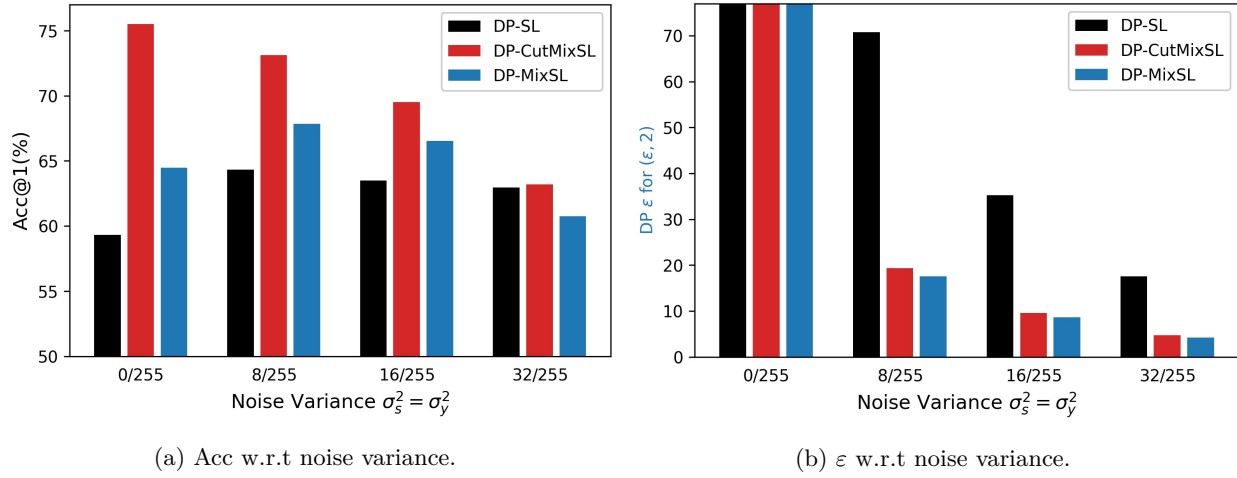

(a) Acc w.r.t noise variance.

(b) $\varepsilon$ w.r.t noise variance.

Figure 11: Accuracy and $\varepsilon$ of DP-CutMixSL, DP-MixSL, and DP-SL w.r.t noise variance.

## E  Proof of Theorem 1

**DP-SL Analysis.**  We first demonstrate the RDP guarantee of DP-SL as follows:

**Proposition 1.** *For all integer $\alpha \geq 2$, $\mathcal{M}_1$ is $(\alpha, \epsilon_1(\alpha))$-RDP, where*

$$\epsilon_1(\alpha) = \frac{\alpha \Delta^2 D_s}{2\sigma_s^2} + \frac{\alpha D_y}{2\sigma_y^2}. \tag{14}$$

Proof. *Starting from the Definition 2 and the Rényi divergence formula with multi-variate Gaussian distributions (Gil et al., 2013), the RDP bound of $\mathcal{M}_1$, denoted by $\epsilon_1(\alpha)$, can be expressed as:*

$$\epsilon_1(\alpha) = \sup_{D,D'} D_\alpha(\mathcal{M}_1(D)\|\mathcal{M}_1(D')) = \sup_{D,D'} \underbrace{\frac{\alpha \cdot \|\mu_s^D - \mu_s^{D'}\|^2}{2\sigma_s^2}}_{=\epsilon_{1,s}(\alpha)} + \underbrace{\frac{\alpha \cdot \|\mu_y^D - \mu_y^{D'}\|^2}{2\sigma_y^2}}_{=\epsilon_{1,y}(\alpha)}, \tag{15}$$

*since $\bar{\mathbf{s}}_i \sim \mathcal{N}(\mu_s^D, \sigma_s^2 I_{D_s})$ and $\bar{\mathbf{y}}_i \sim \mathcal{N}(\mu_y^D, \sigma_y^2 I_{D_y})$, where $\mu_s^D$ and $\mu_y^D$ indicate the average of smashed data and that of label, belonging to dataset $D$. It is noteworthy that $\epsilon_1(\alpha)$ is represented as the sum of RDP bound for smashed data $\epsilon_{1,s}(\alpha)$ and RDP bound for label $\epsilon_{1,y}(\alpha)$ via the sequential composition rule, aiming to induce smashed data-label pairwise RDP bound.*

*Here, by using assumptions about the pixel-wise upper bound of the smashed data and labels ($\mathbf{s}_i \in [0, \Delta]^{D_s}$ and $\mathbf{y}_i \in [0, 1]^{D_y}$), we have:*

$$\|\mu_s^D - \mu_s^{D'}\|^2 \leq \Delta^2 \cdot D_s, \qquad \|\mu_y^D - \mu_y^{D'}\|^2 \leq 1^2 \cdot D_y = D_y. \tag{16}$$

*Combining Equation 15 and Equation 16 concludes the proof.* ∎

**DP-MixSL Analysis.**  We also can present the privacy guarantee of DP-MixSL as belows:

**Proposition 2.** *For all integer $\alpha \geq 2$, $\mathcal{M}_2$ is $(\alpha, \epsilon_2(\alpha))$-RDP, where*

$$\epsilon_2(\alpha) = (\max_{i \in \mathbb{C}} \lambda_i)^2 (\frac{\alpha \Delta^2 D_s}{2\sigma_s^2} + \frac{\alpha D_y}{2\sigma_y^2}). \tag{17}$$

Proof. *Consider the output of DP-MixSL where $n$ smashed data and labels are mixed up, and their pixel-wise upper bound and dimension. Then, for two adjacent datasets $D$ and $D'$ (i.e., only $i'$-th elements are different, $1 \leq i' \leq n$), we have:*

$$\|\mu_s^D - \mu_s^{D'}\|^2 \leq (\lambda_{i'}\Delta)^2 D_s, \qquad \|\mu_y^D - \mu_y^{D'}\|^2 \leq \lambda_{i'}^2 D_y. \tag{18}$$

*Here, Equation 18 is maximized when $\lambda_{i'}$ is the maximum value of $\lambda_i$ for all $i$. Expressing this is as follows:*

$$(\lambda_{i'}\Delta)^2 D_s \leq (\max_{i \in \mathbb{C}} \lambda_i \cdot \Delta)^2 D_s, \qquad \lambda_{i'}^2 D_y \leq (\max_{i \in \mathbb{C}} \lambda_i)^2 D_y. \tag{19}$$

*Recalling the Rényi divergence formula and combining it with Equation 19 completes the proof.* ∎

**DP-CutMixSL Analysis.**  Lastly, the following privacy guarantee of DP-CutMixSL is induced:

**Proposition 3.** *For all integer $\alpha \geq 2$, $\mathcal{M}_3$ is $(\delta, \epsilon_3(\alpha))$-RDP, where*

$$\epsilon_3(\alpha) = (\max_{i \in \mathbb{C}} \lambda_i) \cdot (\frac{\alpha \Delta^2 D_s}{2\sigma_s^2} + (\max_{i \in \mathbb{C}} \lambda_i)\frac{\alpha D_y}{2\sigma_y^2}). \tag{20}$$

Proof. *If we consider the mean of $\tilde{\mathbf{s}}$ for two adjacent datasets $D$ and $D'$, where only the $i'$-th element is different, $N_{i'}P^2C$ pixels among the total $D_s$ pixels are different and the rest are identical. At this time, considering the upper bound of the pixel-level, the following inequality is established:*

$$\|\mu_s^D - \mu_s^{D'}\|^2 \leq (N_{i'}P^2C)\Delta^2 = (\lambda_{i'}NP^2C)\Delta^2 = \lambda_{i'}\Delta^2 D_s. \tag{21}$$

*When $\lambda_{i'} = \max_{i \in \mathbb{C}} \lambda_i$, that is to say, $N_{i'} = \max_{i \in \mathbb{C}} N_i$, Equation 21 is maximized as follows:*

$$\|\mu_s^D - \mu_s^{D'}\|^2 \leq (\max_{i \in \mathbb{C}} \lambda_i)\Delta^2 D_s \tag{22}$$

$$= (\max_{i \in \mathbb{C}} N_i)P^2C\Delta^2. \tag{23}$$

*Recall the Rényi divergence formula, and substitute Equation 23 to obtain a privacy guarantee for DP-CutMix smashed data. Since $\mathcal{M}_3$ is identical to $\mathcal{M}_2$ in terms of labels, the proof is completed by applying this to the RDP sequential composition rule together with $\mathcal{M}_2$'s label privacy guarantee.* ∎

## F   Accuracy comparison of regularizations applied to the input layer

Table 6: Top-1 accuracy of SL-based methods for various datasets and models.

| Method | Models w/ CIFAR-10 | | | Models w/ Fashion-MNIST | | |
|---|---|---|---|---|---|---|
| | ViT-Tiny | PiT-Tiny | VGG-16 | ViT-Tiny | PiT-Tiny | VGG-16 |
| PSL w. Mixup | 74.36 | 37.21 | 66.08 | 89.86 | 87.62 | 89.70 |
| PSL w. Random Cutout | 22.03 | 45.19 | 66.32 | 88.65 | 88.51 | 89.62 |
| PSL w. Vanilla CutMix | 73.02 | 33.54 | 47.69 | 88.72 | 88.37 | 90.02 |
| CutMixSL (proposed) | **75.06** | **53.93** | **67.43** | **89.91** | **89.53** | **90.32** |

## G   Impact of Shuffling

| Method | ViT-Tiny | PiT-Tiny | VGG-16 |
|---|---|---|---|
| PSL | 57.05 | 52.28 | 62.62 |
| **PSL w. Shuffling** | 54.47 | 45.67 | 49.82 |
| PSL w. Mixup | 71.02 | 65.92 | **74.43** |
| PSL w. Random Cutout | 65.03 | 60.87 | 67.06 |
| PSL w. Random CutMix | **75.55** | **73.19** | 72.23 |
| **PSL w. Random CutMix & Shuffling** | 72.78 | 57.59 | 33.50 |

Table 7: Impact of patch-wise shuffling on model accuracy.

| Method | Train Dataset(10%) | Train Dataset(100%) |
|---|---|---|
| PSL | 0.0091 | 0.0056 |
| **PSL w. Shuffling** | 0.0672 | 0.0595 |
| PSL w. Mixup | 0.0402 | 0.0351 |
| PSL w. Random Cutout | 0.0920 | 0.0829 |
| PSL w. Random CutMix | 0.0458 | 0.0434 |
| **PSL w. Random CutMix & Shuffling** | **0.1233** | **0.1250** |

Table 8: Impact of patch-wise shuffling on reconstruction MSE loss.

