# OpenReview forum: "Privacy-Preserving Split Learning with Vision Transformers using Patch-Wise Random and Noisy CutMix"
_TMLR — Accepted by TMLR_

### Review · Reviewer_eXdQ · 2024-05-28

**Summary Of Contributions:**

The paper addresses an important problem in computer vision and proposes a novel framework, DP-CutMixSL, to mitigate data privacy risks in split learning with vision transformers. The paper provides a clear story and the proposed framework is well motivated. However, I find methodology section a bit lacking. Additionally, the paper should discuss the limitations and potential drawbacks of the proposed approach. Therefore, further improvements are needed before the paper can be considered for publication in a peer-reviewed journal.

**Audience:**

No

**Claims And Evidence:**

No

**Requested Changes:**

Addressing the weaknesses mentioned above

**Strengths And Weaknesses:**

### **strengths**
The paper works on an important problem; proposes an interesting approach.

### **weaknesses**
There should be more detailed explanations of the technical aspects of the proposed framework, such as the implementation of the mixer and the process of injecting noise into the smashed data. Also, the limitations of the proposed framework are not discussed. It would be helpful to address potential challenges or drawbacks of the DP-CutMixSL approach.

---

> ### Author Response · Authors · 2024-06-25
> **Response to Reviewer eXdQ**
>
> We thank the reviewer for providing valuable comments that can improve our manuscript. We've incorporated your comments into the revised manuscript as follows:
>
> -	Regarding technical details:
> As you suggested, we have added more details about the implementation of the mixer in section 4, especially in the case of homomorphic encryption. In section 6, we have added additional explanations about the implementation of noise in one-hot encoded labels and added additional clarifications about the behavior of the mixer for the case of n=10 and k>2.
>
> -	Regarding Potential challenges
> As per your suggestion, we have added a discussion of the possible connection to patch-wise shuffling as utilized in existing studies in the conclusion, and create a simple result check in appendix G.

---

### Review · Reviewer_Zgri · 2024-06-09

**Summary Of Contributions:**

This paper proposes a method for privacy-preserving split learning (SL) for training vision transformers (VIT) in computer vision tasks. Their method, coined DP-CutMixSL, adds gaussian noise to clients' smashed data and mixes randomly subsampled patches of clients' smashed data, before sending the noisy data to the server. A differential privacy (DP) guarantee is proven for DP-CutMixSL. Experiments show that DP-CutMixSL offers accuracy and robustness benefits over baselines.

**Audience:**

Yes

**Broader Impact Concerns:**

None.

**Claims And Evidence:**

Yes

**Requested Changes:**

Please see above.

**Strengths And Weaknesses:**

STRENGTHS:

- Practically important problem: an effective, privacy-preserving method for training VIT could be useful in many CV applications
- Section 3 is relatively clear
- Formal DP guarantees are provided for their method.
- Experiments are used to support the efficacy of their method.

WEAKNESSES:

- Presentation and writing is the main weakness: the writing is unclear in places. One general concern is that too many technical details and too much jargon is used early in the paper before defining these terms and outlining the approach at a high level. Other presentation issues:

-- Abstract: too much technical details; second sentence is confusing and run-on; "by analysis, we prove..." sentence is confusing and has strange terminology; "DP-CutMixSL improves accuracy..." compared to what?

-- Use \citet and \citep appropriately

-- terms like "cut-layer representation" and "smashed data" are used before they are defined

-- Figure 1 is too busy: it is not clear what CutMixSL is doing from the figure. Perhaps taking out the attacks would help.

-- Table 1 and Sections 5 & 6 are strange: usually we fix the privacy parameter $\epsilon$ and then write what noise variance $\sigma^2$ is needed to ensure $\epsilon$-(R)DP. An interpretable, clean way to compare two DP mechanisms is to plot accuracy vs. $\epsilon$. However, the paper does things unconventionally and thus it is difficult to interpret the results.

- Related Work: you could discuss related works on differentially private federated learning, such as the papers about "Private federated learning without a trusted server..." by Lowy & Razaviyayn (ICLR 2023 and AISTATS 2023).

---

> ### Author Response · Authors · 2024-06-25
> **Response to Reviewer Zgri**
>
> We appreciate the reviewer’s valuable comments, which have helped improve our manuscript. We have incorporated your suggestions into the revised manuscript as follows:
>
> -	Regarding presentation and writing:
> As you pointed out, we have done our best to improve the presentation of the abstract and introduction. We have changed the citation method for the full text, changed figure 1 and added a description of DP-FL in related works.
>
> -	Regarding accuracy vs. $\varepsilon$ plot:
> Based on your suggestion, we have changed figure 5a. We could still see the higher accuracy and privacy of DP-CutMixSL except for the case \varepsilon=1.

---

### Review · Reviewer_wz2M · 2024-06-16

**Summary Of Contributions:**

The authors studied an important research question:  how we can do privacy-preserving split learning with vision transformers. In response to this research question, a privacy-preserving method based on patch-wise random and noisy CutMix is proposed. The proposed method is evaluated and compared with traditional split learning methods and a theoretical bound based on differential privacy is provided. The proposed method can improve accuracy and robustness to imbalanced data distributions over clients.

**Audience:**

Yes

**Broader Impact Concerns:**

It is recommended to have some discussion about broader impact concerns as they are not seen in the current manuscript.

**Claims And Evidence:**

No

**Requested Changes:**

I would be happy to see this paper published but I think the following changes may need to be gone through before it is qualified for acceptance.

- The abstract includes too much detail about split learning which can be removed to the background and related work. The abstract is supposed to include more general ideas.
- The authors may not need to typically mention FL. The argument of why FL is not suitable is not well-supported. For example, model averaging is not the major reason that FL is not supported in the study of research questions. The authors are encouraged to think about the real difference between FL and SL. For example, vision image transformers are usually of bigger memory usage where we cannot accommodate full model training on the edge.
- It is correct that a ViT usually does not have pooling layers. But a ViT usually has drop out layers which have the similar functionality of skipping activations. Some discussion about drop outs is needed here with the comparison to CNNs.
- It feels that the proposed CutMix method can also be applied to none-patch wise cases. Could the authors provide more rationale about why we need to define such operations on the patch-wise level?
- The resolution of images in CIFAR10 and FashionMNIST is very low, less than 32x32. It is very hard to thoroughly evaluate the success of reconstruction attacks. The authors may need to evaluate over datasets containing images of higher resolution, e.g. CelebA High resolution.
- Only privacy budgets are used as a metric to evaluate the ability to preserve privacy. The authors may need to show the attack success rate in the membership inference attack and the reconstruction performance in the reconstruction attack. Some visualization results are better to understand the effectiveness of privacy preserving methods. For example, moving Figure 10 into the main part.
- What dataset is used to train auxiliary networks in reconstruction attacks?
- The authors need to include important references and have a comparison with existing solutions over privacy-preserving split learning using ViTs.

There are also some minor changes which can be gone through as well:

- The texts in Figure 5 are too small. Readers have to zoom in on every figure.
- The acronym FP for forward propagation is not a common usage.

**Strengths And Weaknesses:**

Thanks for submitting the paper to TMLR and it is a pleasure to see the contribution towards the field of privacy-preserving split learning.

Strengths:
1. The paper studied an important area in privacy-preserving machine learning The method is effective in preserving privacy in split learning with the usage of ViT. The method can effectively defend against three different types of attacks: membership inference attack, reconstruction attack and label inference attack.
2. The authors provided a theoretical privacy guarantee through differential privacy. Both bounds with centralized differential privacy and Renyi-differential privacy are provided
3. The paper is easy to read and flows. Motivation and comparison with conventional models such as CNNs are provided.

Weakness:
1. The paper lacks reference and comparison to important references, which makes it hard to evaluate whether the main argument is well-supported. For example, [1] also proposed a privacy-preserving split learning method based on patch-wise operations, designed for ViTs. [2] studied the privacy-preserving split learning over more broad applications including ViTs for image classification and transformers for text tasks.
2. Though this paper is easy to read and flows, there still exist some organizational issues. For example, why the degradation of ViT over small datasets is related to the high-level idea of this paper. I'm confused as to why authors need to paraphrase such topics. The authors may also need to pay attention to the usage of prepositions in the paper.

[1] Yao, Dixi, Liyao Xiang, Hengyuan Xu, Hangyu Ye, and Yingqi Chen. "Privacy-preserving split learning via patch shuffling over transformers." In 2022 IEEE International Conference on Data Mining (ICDM), pp. 638-647. IEEE, 2022.
[2] Xu, Hengyuan, Liyao Xiang, Hangyu Ye, Dixi Yao, Pengzhi Chu, and Baochun Li. "Permutation Equivariance of Transformers and Its Applications." In Proceedings of the IEEE/CVF Conference on Computer Vision and Pattern Recognition, pp. 5987-5996. 2024.

---

> ### Author Response · Authors · 2024-06-25
> **Response to Reviewer wz2M**
>
> We are grateful for the reviewer’s insightful comments, which have significantly enhanced our manuscript. We have addressed your suggestions in the revised version as follows:
>
> -	Regarding Weakness 1:
> We have carefully reviewed the references you suggested and compared them with our work. The previous work aims to design privacy-preserving SLs without loss of accuracy in single-client situations, while our work aims to design accurate and privacy-preserving parallel SLs in multi-client situations with limited data (as in typical distributed learning situations). The former and latter are strictly different in scope, and we further describe these differences in related works. In addition, the patch shuffling technique, which is representative of the former, can be applied independently to the proposed DP-CutMixSL, and thereby a simple extension experiment is described in Appendix G, and future research directions are described in the conclusion.
>
> -	Regarding Weakness 2:
> In conjunction with our response to weakness 1, we addressed this topic in the sense that limited data in distributed learning scenarios can lead to more severe accuracy degradation, especially in ViT situations.
>
> -	Regarding Writing:
> Based on your comments, we have carefully revised the abstract and introduction. We have also made minor changes such as text size in figures and acronyms.
>
> -	Regarding Dropout Layers:
> We appreciate the point made by review. We have modified the paper to write about the size or spatial information of the smashed data when considering the dropout layer as well as the pooling layer between ViT and CNN.
>
> -	Patch wise vs. none-patch wise:
> We designed the patch-wise behavior as an intuition that the basic unit on which the transformer architecture works is the patch, and there is no particular rationale for it. Instead, we cite references that use patch-wise behavior in ViT to support it. Our intuition was also supported by the accuracy and reconstruction loss comparisons between PSL w. Random CutMix vs. PSL w. Vanilla CutMix, which demonstrated the superiority of patch-wise operation. We have changed our analysis in this regard to make it clearer.
>
> -	Regarding resolution:
> We are preparing figure 6 (of the revised manuscript) for a dataset with a higher resolution than the existing CIFAR-10 dataset. We will update this as soon as possible.
>
> -	Regarding privacy metric:
> As per your suggestion, we added the attack success rate for membership inference attacks and an analysis of it. We also placed figure 6 in the text to visualize the performance of the reconstruction attack.
>
> -	Regarding auxiliary networks:
> We utilized the CIFAR-10 dataset. We have enhanced the description of this.

---

> > ### Author Response · Authors · 2024-06-28
> > **Response to Reviewer wz2M**
> >
> > We have now updated Figure 6 with our experiments with the CelebA dataset. We also noticed that we missed your suggestion about broader impact concerns; if you think it's still needed, please leave a response and we'll add that section to the main text.
> >
> > Thank you for your valuable and constructive comments again.

---

> ### Comment · Reviewer_wz2M · 2024-06-29
> **Response to Rebuttal**
>
> Thanks. I will read the revised manuscript. It is encouraged to have broader impact concerns.

---

> > ### Author Response · Authors · 2024-07-01
> > **Response to Reviewer wz2M**
> >
> > As per your suggestion, we have added a section on broader impact concern in the revised manuscript. Thank you.

---

### Author Response · Authors · 2024-06-25
**Summary of Changes**

Dear Editor and Reviewers,

We would like to express our sincere gratitude to you and the reviewers for your constructive and insightful comments, which have significantly improved the quality of our manuscript. We believe that the revised manuscript reflects all the feedback provided. Below is a summary of our responses to the reviewers’ comments:

1. Regarding Scope:
As per reviewer wz2M's suggestion, we reviewed existing work on privacy-preserving SL in ViT. We have clearly added the scope differences from our work to the related works and conclusions. Additionally, as per reviewer eXdQ's comments, we have included a description and initial results of DP-CutMixSL's potential challenges with patch shuffling in the conclusions and Appendix G, respectively.

2. Regarding Technical Details:
In response to comments from reviewers eXdQ and wz2M, we have added more details on the mixer and noise, discussed the dropout layer, and provided settings for the auxiliary network for the reconstruction attack.

3. Regarding Presentation:
In response to reviewers wz2M and Zgri, we have revised the abstract and introduction, reducing unnecessary details and clearly defining terms, as well as complementing related works.

4. Regarding Experiments:
As per reviewer Zgri's suggestion, we updated the plotting of Figure 5a. Additionally, we added the attack success rate for membership inference attacks and visualized reconstruction attacks in the main text, as per reviewer wz2M's comment.

Once again, we thank you and the reviewers for your valuable feedback, which has been instrumental in enhancing our manuscript. We hope that the revised version meets your expectations and look forward to your positive response.

Sincerely,

Anonymous Authors

---

### Decision · Action_Editor_fb9X · 2024-07-31

**Recommendation:** Accept as is

**Comment:**

The results are not super interesting and the problem setting is a little contrived. However, I recommend acceptance because of the correctness of the claims, and the fact that the claims are interesting to a section of the community

**Audience:**

Audience will be the section of the ML community interested in privacy-preserving ML

**Claims And Evidence:**

This paper studies an interesting problem and appears to provide a correct solution with clear evidence. As a resultI recommend acceptance.